# Tempering of cocoa butter and chocolate using minor lipidic components

Jay Chen [1], Saeed M. Ghazani [1], Jarvis A. Stobbs[1,2] & Alejandro G. Marangoni [1✉]

Chocolate manufacture includes a complex tempering procedure to direct the crystallization of cocoa butter towards the formation of fat crystal networks with specific polymorphism, nano- and microstructure, melting behavior, surface gloss and mechanical properties. Here we investigate the effects of adding various minor non-triglyceride lipidic components to refined cocoa butter and chocolate on their physical properties. We discover that addition of saturated phosphatidylcholine and phosphatidylethanolamine to neutralized and bleached cocoa butter or molten and recrystallized commercial chocolate at 0.1% (w/w) levels, followed by rapid cooling to 20 °C in the absence of shear, accelerates crystallization, stabilizes the desirable Form V polymorph and induces the formation of chocolate with an optimal microstructure, surface gloss and mechanical strength. Final chocolate structure and properties are comparable to those of a commercial tempered chocolate. Minor lipidic component addition represents an effective way to engineer chocolate material properties at different length scales, thus simplifying the entire tempering process.

[1] Department of Food Science, University of Guelph, Guelph, ON, Canada. [2] Canadian Light Source Inc., Saskatoon, SK, Canada.
✉email: amarango@uoguelph.ca

Cocoa Butter (CB), the fat extracted from the *Theobroma cacao* tree, composes close to 30% of chocolate, and many important characteristics of high-quality chocolate are highly dependent on the crystalline structure of CB triacylglycerols (TAGs)[1,2]. CB TAGs have very complex crystallization behavior, and at least six distinct polymorphic forms are known to exist. The triclinic Form V, also known as $\beta_2$, is of particular interest to the chocolate-making industry due to its unique physical properties that impart the most preferable chocolate quality characteristics. Chocolate structured by Form V CB has the proper texture, gloss, snap, and melting profile, and exhibits good thermal and bloom stability. However, attaining this form can be challenging or tedious, often demanding very specific tempering processes that include specific cooling rates, target temperatures, and shear.

The effect of TAG composition on CB crystallization behavior and polymorphism has been widely studied[1,3,4]. On the other hand, however, very little work has been carried out on the effects of CB minor components on this crystallization behavior and structure. Although sometimes ambiguous, the term "minor components" refers to non-TAG lipid molecules that are present in CB at levels of ~3% or less, and include monoacylglycerols (MAGs), diacylglycerols (DAGs), free fatty acids (FFAs), and phospholipids[1].

Variability in the content of minor components can be due to geographical origins, post-harvest handling, and refining[5]. There are currently only a few studies that focus on these minor components, but the few that exist suggest that these can have a large impact on crystal structure and material properties[1,6–9]. Previous studies have shown that minor components can either promote or retard crystal nucleation and growth in a concentration-dependent fashion. This can be due to these minor component acting as nucleation sites, which promote crystallization, or potentially as poisons to crystal growth by blocking crystal growth sites[8].

Arruda and Dimick[6] investigated the phospholipid content in CB seed crystals, finding that these seed crystals contained a significantly greater proportion of phospholipids compared to bulk CB crystals. They suggested that phospholipids, being amphipathic molecules, may play a role in the nucleation process of TAGs, attributed to their ability to self-assemble into mesomorphic structures that would provide a surface upon which TAGs could nucleate and grow[6]. Very few studies have focused on the effects of other minor components, such as MAGs. Chaiseri and Dimick suggested that the low concentration of MAGs in CB may indicate that these components have little effect[1]. Similar to phospholipids, Müller and Careglio reported varying effects of FFAs on the crystallization behavior of CB. Their study indicated that FFAs generally retard crystal growth, and the degree of their effect depends on their concentration[7].

This study aims to elucidate the effects of these minor components on CB crystallization and polymorphism and proposes the idea of utilizing these additives to engineer the solid state and microstructure of CB and chocolate. The concentration of these minor components can be controlled through refining or by direct addition. If minor components can be employed effectively to induce the formation of specific CB polymorphs, as well as induce the formation of an optimal chocolate microstructure with desirable properties, the need for tempering chocolate could be reduced or eliminated.

## Results and discussion

**Cocoa butter composition.** Prior to any refining, the fatty-acid composition of the CB used in this study was determined by gas-liquid chromatography. The CB used in this study is mainly composed of stearic acid (37.3%), palmitic acid (25.4%), and oleic acid (33.5%) (Supplementary Table 1), which closely matches typical literature values[4]. The phosphorus content of the unrefined CB is $88 \pm 1$ ppm, or $0.263 \pm 0.004\%$ equivalent phosphatides by weight (Supplementary Table 2), which is similar to the $0.34 \pm 0.07\%$ reported by Arruda and Dimick[6]. After refining, the phospholipids are completely removed from the CB. The FFA content of the unrefined CB is $1.65 \pm 0.01\%$ (as oleic acid)

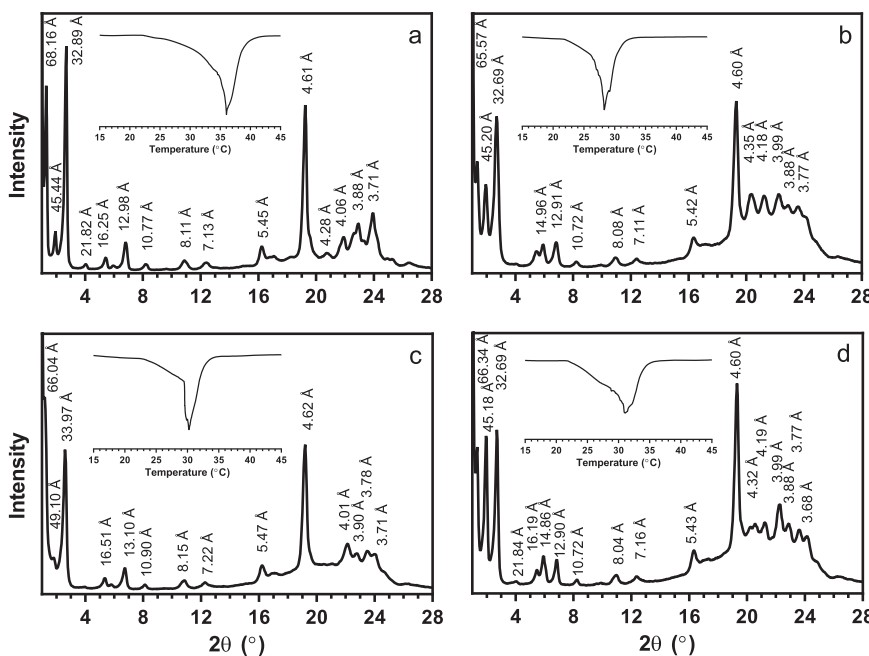

**Fig. 1 Powder X-ray diffraction patterns and differential scanning calorimetric traces (inset) of unrefined native CB, refined CB, and refined CB with added minor lipidic components.** Samples included **a** unrefined native CB, **b** refined CB, **c** refined CB with added 0.1% (w/w) dimyristoylphosphatidylcholine, and **d** refined CB with added 0.1% (w/w) dipalmitoylphosphatidylethanolamine. Samples were crystallized statically at 23 °C for 24 h.

**Table 1 Melting point and enthalpy of fusion of unrefined native CB, refined CB, and refined CB with added minor lipidic components.**

| Sample | Unrefined | Refined | DMPC | DPPE | GMS | GMP | GMO | Stearic | Oleic | Palmitic |
|---|---|---|---|---|---|---|---|---|---|---|
| Melting point (°C) | 36.4 ± 0.2[a] | 28.8 ± 0.2[b] | 30.8 ± 0.3[c,d] | 31.7 ± 0.7[c,d] | 29.8 ± 0.5[b,c] | 30.4 ± 0.5[b,c,d] | 32.2 ± 0.2[d,e] | 34.7 ± 0.3[a,f] | 34.0 ± 0.2[e,f] | 34.2 ± 0.2[f] |
| Enthalpy of fusion (J/g) | 143.9 ± 0.7[a] | 110.9 ± 0.9[b] | 105.5 ± 1.3[b] | 106.2 ± 4.0[b] | 106.6 ± 1.4[b] | 105.0 ± 3.0[b] | 104.4 ± 0.5[b] | 145.0 ± 0.3[a] | 141.4 ± 0.9[a] | 140.4 ± 2.9[a] |

Samples were crystallized statically at 23 °C for 24 h. Values represent the means and standard errors of three replicates. Values with the same superscript letter within a row are not significantly different (*P* > 0.05).

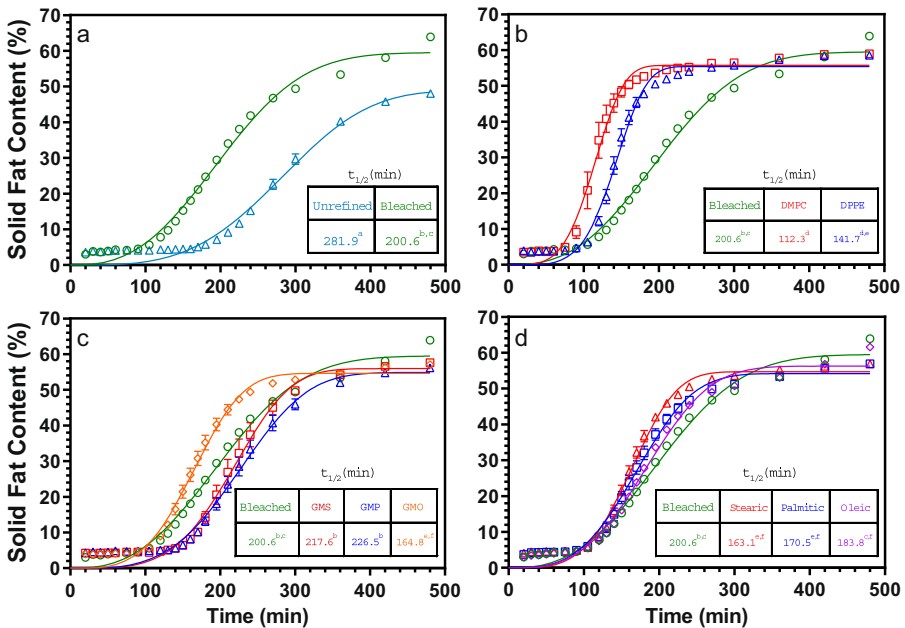

**Fig. 2 Static crystallization growth curve of unrefined native CB, refined CB, and refined CB with added minor lipidic components crystallized at 23 °C (symbols) and corresponding fits to the Avrami model (solid lines).** The inset table indicates the half-times of crystallization derived form model fits. Refining CB accelerates its crystallization (**a**), as does 0.1% (w/w) phospholipid addition (**b**). Addition of 0.5% (w/w) saturated monoacylglycerols, glycerol monostearate, and glycerol monopalmitate, do not significantly affect crystallization kinetics, while addition of the unsaturated glycerol monooleate does (**c**). Free fatty-acid addition to refined CB at 0.5% (w/w) levels accelerates crystallization for stearic, palmitic, and oleic acids equally (**d**). Values represent means and standard errors of three replicates and values with the same superscript letter are not significantly different (*P* > 0.05).

(Supplementary Table 3), which complies with the requirement of having <1.75% FFAs as oleic acid in Codex standards. This is reduced to 0.06% after neutralization and bleaching.

**Crystal structure and polymorphism.** Powder X-ray diffraction is the most unambiguous method for determining CB TAG polymorphism. In this study, CB samples were melted at 80 °C for 30 min to clear the memory of crystallization and then crystallized statically at 23 °C in an incubator for 1 day. Form IV displays wide-angle reflections corresponding to short spacings of 4.35, 4.15, 3.97, and 3.81 Å, Form V has short spacings of 4.58, 3.98, 3.87, 3.75, and 3.67 Å, while Form VI has short spacings of 4.59, 3.86, and 3.70 Å[10]. Unrefined CB displays reflections corresponding to short spacings of 4.61, 3.88, and 3.71 Å, which is characteristic of Form VI, matching literature values closely (Fig. 1; Supplementary Table 4)[10]. Additionally, the small-angle reflections corresponding to long spacings of 68.16 and 32.89 Å further suggest a Form VI polymorphism and a 3 L structure (polytype)[11]. CB TAGs in Forms V and VI stack as lamellae in the [001] direction with a dimension, which is proportional to three fatty-acid chain lengths, hence the nomenclature "3 L". Other polymorphic forms stack as 2 L structures, the more common polytypic arrangement[12].

In contrast, the refined CB sample displays a pattern that resembles a blend of Forms IV and V. Short spacings of 4.35 and 4.18 Å agree with literature values for Form IV, while 4.60, 3.99, 3.88, and 3.77 Å match those of Form V[11]. Furthermore, the long spacings include both 2 L (45.20 Å) and 3 L (65.57 and 32.69 Å) arrangements, indicative of the presence of Forms IV and V, respectively[11]. Thus, it appears that the removal of minor components in CB through refining drastically changes the crystal structure of CB.

The wide-angle region of powder XRD spectra of refined CB with minor components added back is generally characteristic of Form V, or a blend of Forms IV and V (Fig. 1c, d and Supplementary Fig. 1 (a–f)). In the small-angle region, diffraction patterns suggest that the 3 L structure is present in all samples. Long spacings in the range 64–68 Å correspond to the reflections of the (001) crystal plane. Higher-order reflections, up to ones corresponding to the (009) plane of this lamellar crystal, are also evident. However, there is also varying amounts of the 2 L structure present in the various samples, specifically Form IV, which imparts undesirable properties and stability to CB and chocolate, including a low melting point and excessive softness[12]. The DMPC, GMP, GMO, and FFA samples have little to no 2 L structure present, indicated by the lack of a (001) plane reflection corresponding to ~45 Å. In contrast, DPPE and GMS CB samples

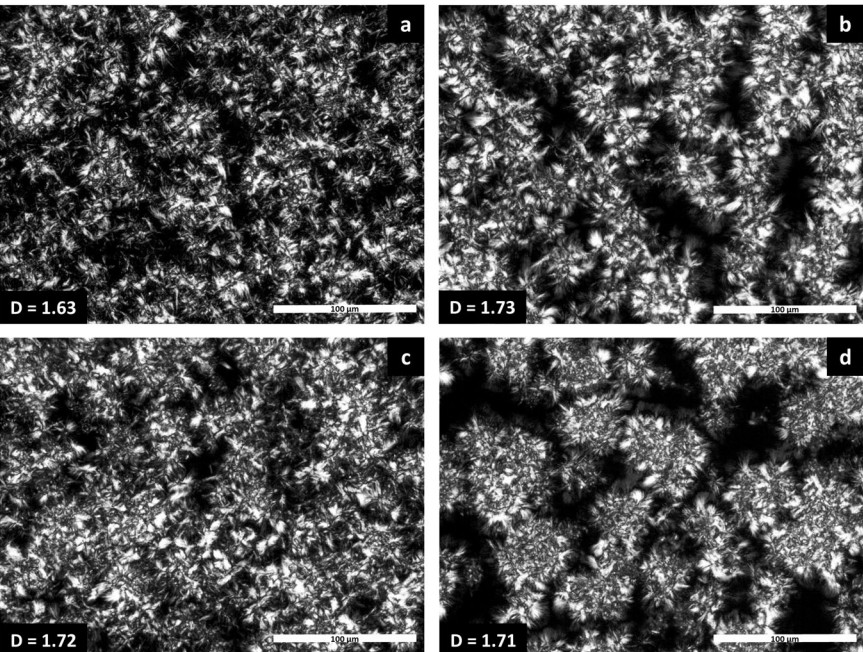

**Fig. 3 Polarized light microscopy images of unrefined native CB, refined CB, and refined CB with and without added minor lipidic components.** Samples included **a** unrefined native CB, **b** refined CB, **c** refined CB with added 0.1% (w/w) dimyristoylphosphatidylcholine, and **d** refined CB with added 0.1% (w/w) dipalmitoylphosphatidylethanolamine. Samples were crystallized statically at 23 °C for 24 h. Intensity and contrast were auto-levelled.

display a strong 2 L reflection, suggesting a mixture of 2 L and 3 L structures.

**Melting point and enthalpy of fusion**. The melting points and enthalpies of fusion determined by DSC were compared across the CB samples and are shown in Table 1. Example DSC curves are presented for each sample in Fig. 1 and Supplementary Fig. 2. Wille and Lutton reported the melting temperatures for Forms I through VI as 17.3, 23.3. 25.5, 27.5, 33.9, and 36.3 °C[10], while Lovegren et al.[13] reported lower melting temperatures for the same six polymorphs as 13, 20, 23, 25, 30, and 33.5 °C. The melting point of the unrefined CB used in our study closely matches that of Form VI, as reported by Willie and Lutton[10]. However, once the CB is refined, the melting point drops to 28.8 °C, indicative of Form IV. After adding specific minor components back into the refined CB, the melting temperature increases again. Phospholipid and MAG addition to refined CB yield samples with melting points in the range of 29.8–32.2 °C, suggesting that the CB crystals are in transition from Form IV to Form V, as reported by Wille and Lutton[10]. Finally, when FFAs are added to refined CB, the melting temperature increases back to about 34 °C, which may indicate a mixture of Forms V and VI, as it is also similar to Form VI's melting point of 33.5 °C[13]. These results are all in agreement with the XRD data.

The enthalpies of fusion follow a similar trend, with a higher enthalpy of fusion (143.9 J/g) for unrefined CB, which decreases significantly after refining (110.9 J/g). Again, the phospholipids and MAGs have a much smaller effect, if any, compared to the FFAs. With the FFA samples, the enthalpies of fusion are similar to that of the unrefined CB. Guthrie reported an enthalpy of 147 J/g for Form V CB, which decreased for less stable forms[14].

**Crystallization kinetics**. The crystallization kinetics of the CB samples were quantified using half-times derived from the Avrami model fitted to the SFC-temperature data, as shown in Fig. 2, with the calculated Avrami constants shown in Supplementary Table 5[15].

Unrefined CB has the longest half-time of crystallization at 281.9 min, which decreases to 202.3 min after refining. Statistically significant decreases ($P < 0.05$) in the half time of crystallization of unrefined CB are observed after addition of DMPC, DPPE, GMO, and all free fatty acids tested, including palmitic, stearic, and oleic acids. Chaiseri and Dimick reported that different phospholipids had different effects on crystallization rate, with certain phospholipids such as lysophosphatidylcholine and phosphatidylinositol retarding nucleation, while others such as phosphatidylcholine promoting nucleation. They also highlighted that removing phospholipids through refining decreased the crystallization rate, indicating that phospholids may be necessary as initial crystallization seeds in CB[1]. This agrees with the data presented in this study, with the addition of phospholipids increasing crystallization rate. Arruda and Dimick reported a much higher relative concentration of phospholipids in seed crystals than in bulk CB, with the majority being phosphatidylethanolamine and phosphatidylcholine[6]. Thus, it appears that the added phospholipids may aid in the nucleation of CB TAGs.

The FFA and GMO CB samples have a slightly shorter half-time of crystallization relative to the refined CB sample, but the effects of palmitic and oleic acid are not statistically significant ($P > 0.05$). Müller and Careglio focused their study specifically on FFAs, and reported slower crystal growth of unrefined CB with added stearic, palmitic, and oleic acids[7]. However, we do not observe this in our study. Chaiseri and Dimick reported that saturated free fatty acids increased the crystallization rate in the early stages of CB crystallization[1].

Few studies have been conducted on the effect of MAGs on CB crystallization. Chaiseri and Dimick suggested that their effects are small due to their low concentration in CB[1]. In our study, GMS and GMP, both saturated monoglycerides, do not

significantly affect CB crystallization kinetics. Surprisingly, the unsaturated monoglyceride GMO enhances CB crystallization.

**Polarized light microscopy**. Light microscopy revealed subtle differences in the microstructure of the different CB samples (Fig. 3 and Supplementary Fig. 3). Although all images display feather-like spherulitic structures, the unrefined sample has smaller spherulites. DPPE addition, interestingly, leads to the formation of larger spherulites with a different texture than upon DMPC addition. The spherulites of the refined and DPPE samples display granular structures in the centre of the spherulites and feather-like crystallites in the periphery. The MAG samples all exhibit similar crystal morphologies, with both granular structures and small feather-like spherulites present. This is similar to the stearic and palmitic acid samples, though the FFA samples display a more granular structure. It appears that the MAG and FFA samples, excluding the oleic acid sample, revert the CB to the morphology of the unrefined sample. However, the oleic acid sample differs greatly from the rest, displaying large spherulites analogous to the refined CB sample.

An aspect that became clear during the microscopy studies is that the spatial distribution of mass of CB is affected by the refining process and subsequent addition of minor lipidic components. One way to quantify this spatial distribution of mass is by determining the box-counting fractal dimension ($D_{box}$) of the images. A high fractal dimension is associated with a greater fill of the available embedding space with crystalline mass, as well as a higher order in the distribution of this crystalline mass. The box-counting fractal dimension provides a value that quantifies the level of space-filling, as well as self-similarity within a structure, which can be related to the nucleation process of CB. Supplementary Table 6 lists these values. The unrefined and stearic acid samples have the lowest average fractal dimension, with $D_{box} = 1.63$ and 1.62, respectively. In contrast, the sample with the highest fractal dimension is the refined sample at $D_{box} = 1.75$, followed by the phospholipid samples at ~1.72. Samples with a higher fractal dimension are expected to have originated from a higher nucleation rate, as this leads to more homogenous growth across an area. On the other hand, slower nucleation rates would result in more heterogeneous growth, leading to a lower fractal dimension for the resulting mass distribution [16].

Interestingly, here we also discovered a statistically significant ($r = 0.77$, $P < 0.01$) inverse correlation between the melting temperature of the samples and their fractal dimension, as shown in Fig. 4. This may be related to the polymorphism of each sample and its corresponding nucleation process. For the unrefined

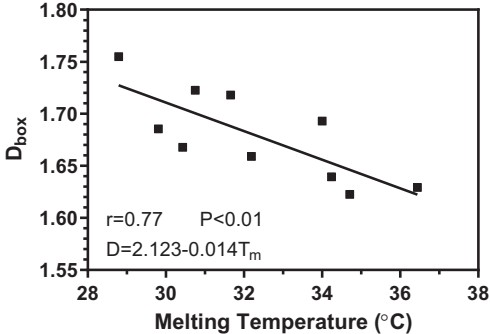

**Fig. 4 Global relationship between the box-counting fractal dimension ($D_{box}$) of polarized light microscopy images of CB samples and their corresponding melting temperature ($T_m$).** Fractal dimension values plotted represent the means of six images, while melting temperature values plotted represent the means of three replicates.

sample, a high melting temperature of 36.4 °C indicates the presence of the Form VI polymorph. The higher free energy of nucleation for this most stable polymorph results in low nucleation rates, leading to a more heterogeneous mass distribution and a lower fractal dimension. Conversely, the lower melting temperature of the refined sample suggests a mixture of Forms IV and V, which are metastable polymorphs. These polymorphic forms have greater rates of nucleation, resulting in a more homogeneous mass distribution and a higher fractal dimension [16].

**Mechanical properties**. The hardness of both CB and chocolate samples containing some of the minor components was determined by 3-point bending, and the elastic bending modulus calculated. The elastic bending moduli ($E_B$) of unrefined CB, refined CB, and DMPC CB are 44.1 ± 3.4, 35.4 ± 4.4, and 39.0 ± 3.7 N/mm$^2$, respectively. There is no statistically significant difference between these values ($P > 0.05$). The melting temperatures for these samples are the same as well, with the average being 34.3 ± 0.1 °C.

For the chocolate samples, the addition of phospholipids to molten commercial chocolate (90% cocoa solids) allows the chocolate to maintain hardness without complex tempering procedures, such as crystallization at 5 °C for 1 h and subsequent storage at 20 °C for 4 days. Other trials with static crystallization at 23 and 26 °C yield chocolates with undesirable mechanical and surface properties, suggesting that the use of phospholipids to promote proper tempering still requires some optimization of crystallization temperatures. DMPC and DPPE were added chocolate at 0.1% (w/w) levels, and their properties were compared to commercial chocolate both directly from the package and recrystallized under the same conditions as the doped samples (Table 2). The $E_B$ is similar between the commercial chocolate and the phospholipid-containing samples, but is significantly lower for the recrystallized commercial chocolate ($P < 0.05$). When the chocolate is melted and recrystallized without any additive addition, it becomes much softer. A greater $E_B$ indicates a stiffer chocolate, relating to the property of snap, which is desirable in chocolate. However, DSC analysis revealed that all the chocolate samples have similar melting temperatures, all indicative of Form V CB crystals. XRD patterns further confirm the presence of Form V CB crystals in all samples (Fig. 5). Thus, it appears that the polymorphism of CB in chocolate does not wholly determine its mechanical properties, and these mechanical properties may be attributed to the microstructure instead.

Proper chocolate tempering typically focuses on the melting behavior of the chocolate, in other words, the polymorphism of the chocolate's CB triacylglycerol crystals [2,3]. However, there is little consideration for the microstructure and mechanical properties when determining whether chocolate has been properly tempered or not. Here we show that addition of DMPC and DPPE helps achieve the desired hardness and fracture properties in chocolate, without directly impacting the polymorphism. This likely results from changes in the microstructure of the underlying CB fat crystal network, which arises from the phospholipids affecting the nucleation behavior of the CB.

**Surface reflectance**. Proper tempering of chocolate results in a smooth surface with high surface gloss [17]. Bloom, for example, is often characterized by the appearance of a grayish white film on the surface of chocolate, attributed to the uncontrolled polymorphic transformation from Form V to Form VI [18]. This white appearance is due to the presence of large surface crystals, which scatter light impinging upon them [19]. This can be quantified using the Whiteness Index (WI), calculated from RGB surface color

**Table 2 Elastic bending modulus ($E_B$), melting temperature, and whiteness index (WI) of commercial chocolate samples with added phospholipids.**

| Sample | Commercial | DMPC | DPPE | Recrystallized commercial |
|---|---|---|---|---|
| $E_B$ | 74.7 ± 13.2[a] | 75.6 ± 8.3[a] | 65.7 ± 8.4[a,c] | 36.6 ± 4.2[b,c] |
| Melting temperature (°C) | 33.3 ± 0.5[a] | 34.4 ± 0.2[a] | 34.0 ± 0.1[a] | 33.6 ± 0.3[a] |
| WI | 38.9 ± 1.6[a,b] | 32.2 ± 3.3[a,*] | 40.0 ± 2.8[a,b,*] | 43.6 ± 2.6[b] |

Samples included fresh commercial chocolate, molten and recrystallized commercial chocolate with added 0.1% (w/w) dimyristoylphosphatidylcholine (DMPC), molten and recrystallized commercial chocolate with added 0.1% (w/w) dipalmitoylphosphatidylethanolamine (DPPE), molten and recrystallized commercial chocolate. The molten and recrystallized samples were crystallized at 5 °C for 1 h and stored at 20 °C for 4 days. Values represent the means and standard errors for 4–16 replicates for $E_B$ and WI and for 3 replicates for melting temperature. Values with the same superscript letter within a row are not significantly different ($P > 0.05$).
*There is a significant difference between these values at $P < 0.1$.

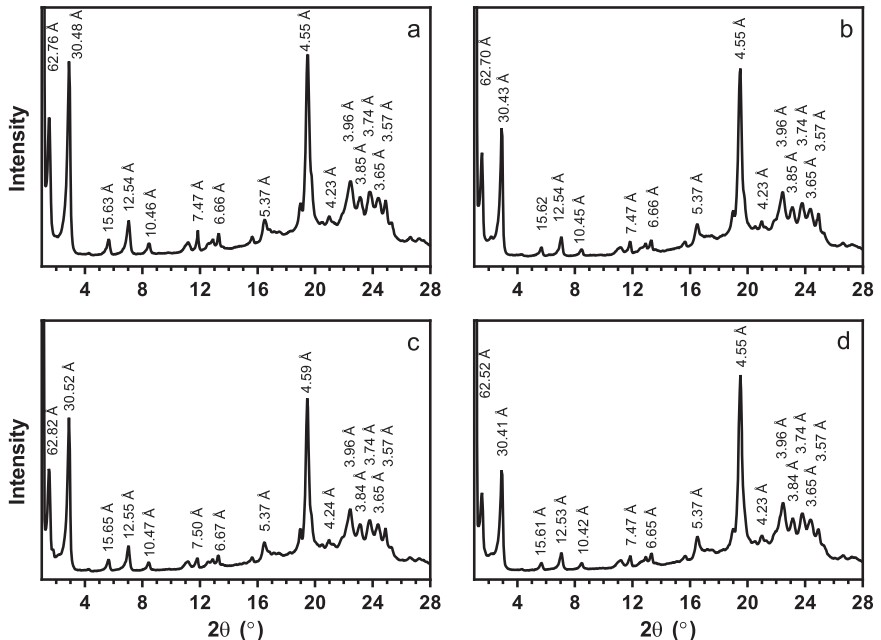

**Fig. 5 Powder X-ray diffraction patterns for commercial chocolate with added minor lipidic components.** Chocolate samples were molten, blended with the minor components and crystallized at 5 °C for 1 h, followed by 20 °C for 4 days. The **a** commercial chocolate, **b** molten and recrystallized commercial chocolate, **c** molten and recrystallized commercial chocolate with added 0.1% (w/w) dimyristoylphosphatidylcholine, and **d** molten and recrystallized commercial chocolate with added 0.1% (w/w) dipalmitoylphosphatidylethanolamine, all display powder X-ray diffraction patterns characteristic of well-tempered $\beta_2$, triclinic, Form V cocoa butter crystals. This crystal form displays a strong reflection corresponding to a short spacing of ~4.6 Å in the wide-angle diffraction region, and characteristic wide-angle reflections in the 2θ region 20–25°. In the small-angle diffraction region, only reflections from the 3 L (001) plane corresponding to ~62–63-Å were observed. The absence of reflections from (001) plane of the 2 L polytype indicates the formation of the optimal polytypic form in the cocoa butter in all samples. Reflections at higher angles in the small-angle region correspond to higher-order reflections for these two crystal planes.

measurements[17]. A higher WI indicates a greater level of fat bloom on the surface of the chocolate, or may also indicate a rougher surface. The WI for each of the chocolate samples is shown in Table 2. The commercial chocolate and the DMPC chocolate sample have similar WI, suggesting a properly tempered chocolate for the DMPC sample. Moreover, a statistically significant difference is evident between the DMPC chocolate sample and the recrystallized commercial chocolate sample ($P < 0.05$), which was not tempered properly. The lower WI of the DMPC chocolate sample suggests that the addition of this phospholipid aided in the stability of the chocolate, resulting in less fat bloom and/or a smoother surface compared to the recrystallized commercial chocolate sample. The effect of DPPE addition to molten and recrystallized chocolate is not as clear since the WI is not statistically different from the commercial, DMPC or molten and recrystallized chocolates.

**Chocolate microstructure by synchrotron-based micro-computed tomography (SR-μCT).** Synchrotron μCT analysis revealed similarities between chocolate microstructure among all samples imaged at a <450 μm length scale. CT voxel intensity is directly proportional to material density. The number of voxels with a given intensity is then representative of the number of volume fractions with a given density in the analyzed volume. The molten and recrystallized commercial chocolate sample (Lindt-R) displays the largest difference between samples, with a narrowing and shift of the histogram compared to the fresh commercial chocolate (Lindt-F), molten and recrystallized commercial chocolate with added DMPC, and molten and recrystallized commercial chocolate with added DPPE. This indicates a difference in material structure upon recrystallization. This is visually represented in volume renderings where the most notable difference is a reduction in the green intensity for Lindt-R,

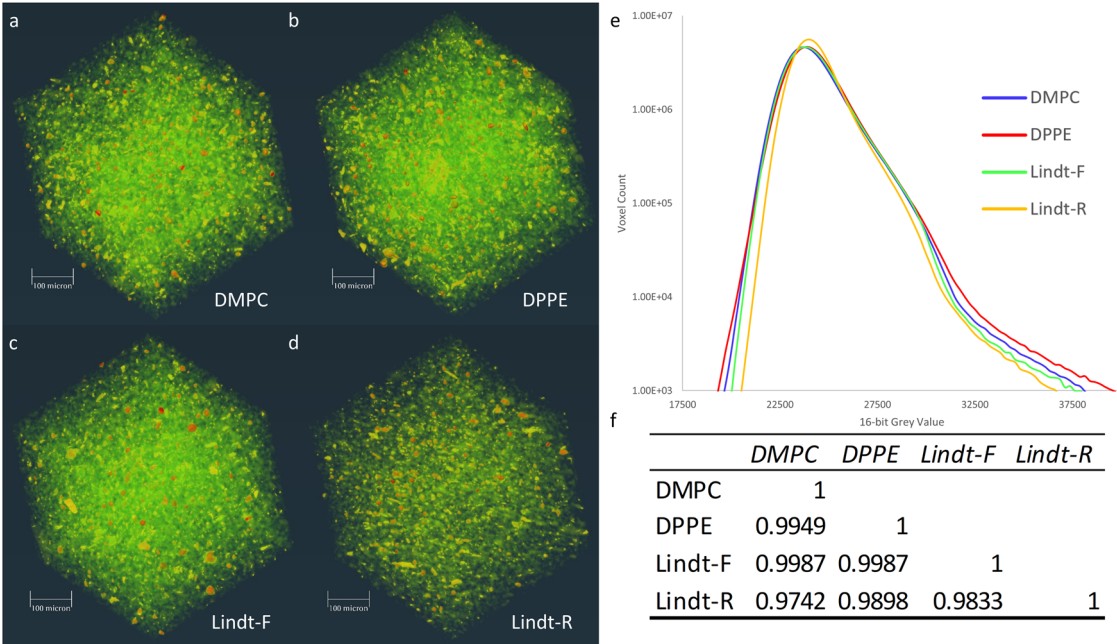

**Fig. 6 Volume renderings of reconstructed slices, plotted histograms, and correlation matrix of SR-μCT data.** Samples included **a** molten and recrystallized commercial chocolate with added 0.1% (w/w) dimyristoylphosphatidylcholine (DMPC), Supplementary Movie 1; **b** molten and recrystallized commercial chocolate with added 0.1% (w/w) dipalmitoylphosphatidylethanolamine (DPPE), Supplementary Movie 2; **c** fresh commercial chocolate (Lindt-F), Supplementary Movie 3; **d** molten and recrystallized commercial chocolate (Lindt-R), Supplementary Movie 4; **e** plotted histograms, and **f** correlation matrix.

compared to the Lindt-F, DMPC, and DPPE samples, as seen in Fig. 6. Volume rendering movies of each sample can be seen in the corresponding links in Fig. 6.

A comparison of the histograms shows qualitative differences in the overall density distributions, which make up the sample volumes in the images. A correlation matrix was generated by inputting the voxel count values for each binned 16-bit gray-value and generating a table of correlation coefficients between the four chocolates density distributions. The correlation matrices show a high positive correlation in the material's density distribution regardless of treatment; however, the least variance is observed between DMPC vs. Lindt-F and DPPE vs. Lindt-F. The highest correlation coefficient is found between DMPC and fresh Lindt, and DPPE vs. fresh Lindt samples, with a value of 0.9987. This suggests that the complex microstructure of Lindt chocolate with added DMPC and DPPE and crystallized isothermally, is more similar to that of tempered fresh Lindt chocolate than without these additives.

Overall, we find that the presence of various minor components within CB results in major changes to its polymorphism, crystallization kinetics, and crystal morphology. By refining CB through neutralization and bleaching, the unrefined CB transitioned from Form VI to a blend of Forms IV and V, in addition to displaying faster crystal growth. Although adding MAGs and FFAs back into refined CB results in some small differences, phospholipid addition has the largest impact.

At 0.1% (w/w) levels, DMPC and DPPE both significantly increase the crystal growth rate of refined CB. However, the DPPE sample exhibits significant amounts of the 2 L polytype, while the DMPC sample only displays 3 L polytypism, which is the more stable and preferred nanostructure. When added to a commercial chocolate sample, DMPC and DPPE both template the Form V polymorph with optimal mechanical properties, surface gloss and microstructure. This may allow for the

production of chocolate products without the need for laborious and complex tempering protocols.

Proper tempering is often judged by the resulting melting temperature, a function of the CB polymorphic form[2]. However, as we show in this study, the various chocolate samples were all in Form V as determined by DSC and XRD analysis, but display different mechanical properties. This suggests that there is more to tempering than achieving the Form V polymorph, and important qualities such as gloss and snap may be heavily dependent on surface and bulk microstructure. Thus, DMPC, DPPE, and possibly other saturated phospholipids can potentially be employed as effective additives for engineering the crystallization behavior, polymorphism, nanostructure, and microstructure in CB and chocolate products.

## Methods

**Samples**. Malaysian CB supplied from JB Cocoa Sdn. Bhd. (JB Cocoa, Johor, Malaysia) was used for all CB samples. Lindt Excellence Cocoa 90% dark chocolate (Chocoladefabriken Lindt & Sprüngli AG, Kilchberg, Switzerland) was used for all chocolate samples. Chocolate samples were melted and poured into 35.0 × 19.0 × 5.0 mm molds and allowed to set at 5 °C for an hour before transferring to an incubator at 20 °C to finish crystallizing for 4 days.

**Fatty-acid composition**. For determination of the fatty-acid profile, the CB was analyzed by an Agilent 6890-series gas chromatograph (Agilent Technologies, Inc., Wilmington, DE, USA) with a 7683-series auto-sampler. A 60 m by 0.22 mm internal diameter BPX70 column (SGE Inc. Austin, TX, USA) with a 0.25 μm film thickness was used. The oven temperature increased at 4 °C/min from 110 to 230 °C and then sustained at this temperature for 18 min. The injector was set at 250 °C and operated at 20.1 psi with a flow of 17.7 mL/min. High-purity helium flowed at an average velocity of 25 cm/s as the carrier gas. A flame-ionization detector was set at 255 °C with an air flow of 450 mL/min and a helium flow of 50 mL/min. The GC separation peaks were analyzed using Open LAB software (Agilent Technologies). Fatty-acid composition was determined by comparing the retention times of the peaks to internal standards. The fatty-acid composition of the added MAGs and GMO were also analyzed in the same manner. Complete results can be found here: https://doi.org/10.7910/DVN/SJ8BEQ.

**Phosphorus content**. To determine the amount of phosphorus in the CB before and after the refining process, the AOCS Official Method Ca 12–55 was employed. The phosphorus content in the sample was determined spectrophotometrically and compared to a prepared standard curve. Complete results can be found here: https://doi.org/10.7910/DVN/SJ8BEQ.

**Free fatty-acid content**. Following the method provided by the AOCS Official Method Ca 5a–40, the free fatty-acid content was determined for the samples both before and after neutralization. The free fatty-acid percentage and acid value were calculated as stated in the method. Complete results can be found here: https://doi.org/10.7910/DVN/SJ8BEQ.

**Refining process**. The CB was neutralized for the removal of free fatty acids. Once the CB was heated to 50 °C, a 2.7% treat of 16% sodium hydroxide was added according to the level of free fatty acids as determined previously. After mixing for 20 min, the temperature was raised to 70 °C and the sample was subsequently mixed for another 5 min. Centrifuging to separate the soap layer, the sample was then washed with hot water in a separatory funnel before heating to evaporate excess water. The neutralized CB was subsequently bleached for further refining by melting and heating to 110 °C before 0.5% bentonite bleaching clay was added. After stirring for 25 min at this temperature, the sample was filtered using a Whatman Grade 4 240 mm filter paper.

**Addition of minor components**. After the refining processes, various phospholipids, MAGs, and FFAs were added to the CB. Cocoa butter was melted at 80 °C for 15 min, and the minor components were mixed in by hand until fully dissolved. Chocolate samples were molten in a double boiler at 60 °C and minor components added and mixed by hand until fully dissolved. Minor components were fully incorporated within 5 min of mixing. 1,2-dimyristoyl-sn-glycero-3-phosphocholine (DMPC) (Avanti Polar-Lipids, Alabaster, Alabama, USA) and 1,2-dipalmitoyl-sn-glycero-3-phosphoethanolamine (DPPE) (Avanti Polar-Lipids, Alabaster, Alabama, USA) were each added to refined CB at 0.1% w/w levels. Glyceryl monostearate (GMS) (Alphadim 90 SBK from Caravan Ingredients, Lenexa, Kansas, USA), glyceryl monopalmitate (GMP) (Alphadim 90 PBK from Caravan Ingredients, Lenexa, Kansas, USA), and glyceryl monooleate (GMO) (1-monoolein, Alfa Chemistry, New York, USA) were added at 0.5% w/w levels. Gas chromatography revealed that the GMS was mainly stearic acid (88.3%), the GMP was not only palmitic acid (56.9%) but also had significant amounts of stearic acid (40.5%), and GMO was mainly oleic acid (81.8% oleic acid). Stearic acid, palmitic acid, and oleic acid (Fisher Scientific Company, Ottawa, Ontario, Canada), as free fatty acids, were added at 0.5% w/w levels. DMPC and DPPE were added to the chocolate samples at 0.1% (w/w) levels.

**Solid fat content (SFC) measurements for crystallization kinetics**. The samples were first melted at 80 °C for 30 min to remove any previous crystal memory. After placing the samples in glass pulsed nuclear magnetic resonance (pNMR) tubes (10 mm diameter, 1 mm thickness, and 180 mm height) in a water bath set at 23 °C, solid fat content measurements were taken using a Bruker mq20 Minispec Series PC 120 pNMR spectrometer operating at 20 MHz and 0.47 T (Milton, ON, Canada) at suitable timepoints. Analysis of data was performed using MiniSpec software V2.51 Rev. 00/NT (Bruker Biospin Ltd., Milton, ON, Canada). Samples were measured in triplicate. The Avrami model was used to quantify crystallization kinetics by fitting the model to SFC-time data by nonlinear regression (SFC = $SFC_{max}(1 - e^{-k_A t^n}$), ref. [15]) using GraphPad Prism 5.0 (GraphPad Software, San Diego, CA, USA). The half-life of crystallization was then determined as $t_{1/2} = (\ln 2/k_A)^{-n}$, ref. [15] Complete results can be found here: https://doi.org/10.7910/DVN/SJ8BEQ.

**Powder X-ray diffraction**. Samples were analyzed by X-ray diffraction (Multiflex Powder XRD spectrometer, Rigaku, Tokyo, Japan) to identify the polymorphism and crystal structure. The instrument was operated at 40 kV and 44 mA and was run with a copper X-ray tube with a wavelength of 1.54 Å. Measurements were taken after CB samples were melted at 80 °C for 30 min and crystallized statically at 23 °C in an incubator for 1 day. Scans ranged from 1 to 30° at a scan rate of 0.5°/min. The XRD patterns were analyzed with Jade 9 (Materials Data Inc., Livermore, CA, USA). Complete results can be found here: https://doi.org/10.7910/DVN/SJ8BEQ.

**Differential scanning calorimetry (DSC)**. To determine melting points and enthalpies of fusion for the samples, a TA instrument model Q2000 (TA instruments, Mississauga, ON, Canada) DSC with a cooling system was utilized. Nitrogen was used to purge the system at a flow rate of 18 mL/min. Five to ten milligrams of samples was placed in an aluminum DSC pan with a lid. These were measured against a reference empty hermetically sealed pan. The pan was equilibrated to 5 °C, before the temperature ramped up at 5 °C/min until 50 °C. Measurements were taken in triplicate. Complete results can be found here: https://doi.org/10.7910/DVN/SJ8BEQ

**Polarized light microscopy (PLM)**. To elucidate the microstructure of the different samples, polarized light microscopy was employed. The CB samples were first melted at 80 °C for 30 min to eliminate previous crystallization memory. A small drop of sample was placed on a pretempered microscope slide, and a pre-tempered glass cover slip was first placed on one edge at the side of the sample and lightly dropped to ensure a uniform layer without air bubbles. The prepared slides were placed in an incubator set at 23 °C for 24 h before examining on a microscope. The samples were imaged by PLM on an OMAX optical microscope model M838PL (OMAX Microscope, USA) equipped with a ×40 objective lens. Images were captured using a model A35180U3 digital camera (OMAX Microscope, USA) using Toupview software (ToupTek Photonics, Zhejiang, China). Images were auto-levelled in Adobe Photoshop 2020 (San Jose, CA, USA) and then auto-thresholded to 128 as shown in Supplementary Fig. 4. The images were analyzed in Benoit 1.3 Fractal Analysis System (TruSoft International, Inc., Miami, FL, USA) to determine the box-counting fractal dimension. Six different micrographs per treatment were used in the box-counting analysis. Complete results can be found here: https://doi.org/10.7910/DVN/SJ8BEQ.

**Three-point bending**. Mechanical properties of the CB and chocolate samples were tested using a Model TA.XT2 texture analyzer (Stable Micro Systems, Texture Technologies Corp., Scarsdale, NY, USA) with a three-point bending attachment. The settings were set to compression mode, with a test speed of 1.00 mm/sec and a trigger force of 0.100 N. Samples were balanced on two points 20.0 mm apart with the three-point bending attachment applying a constant force in the middle to break the sample. The peak force (F) required to break the sample and the deformation of the sample at the peak force (d) were recorded. With these values, along with the height (h) and width (b) of the samples, and the separation between the two support points (a), the elastic bending modulus ($E_B$) ($E_B = \frac{Fa^3}{4dbh^3}$) was calculated[20]. Complete results can be found here: https://doi.org/10.7910/DVN/SJ8BEQ.

**Surface reflectance**. The surface color of the chocolate samples was analyzed using the ColorMeter Free app (https://play.google.com/store/apps/details?id=com.vistechprojects.colormeterfree) on an Android cell phone (Samsung Galaxy S7). Guidance provided by "Use Your Smartphone as an "Absorption Spectrophotometer"" (https://www.chemedx.org/blog/use-your-smartphone-absorption-spectrophotometer) was followed. Samples were placed on a black background and photographed, before analyzing to provide RGB values. These values were converted into XYZ values and subsequently CIE L*a*b* values through nonlinear functions using the EasyRGB website (https://www.easyrgb.com/en/convert.php#inputFORM). These values were then used to calculate the Whiteness Index (WI) ($WI = 100 - [(100 - L^*)^2 + a^{*2} + b^{*2}]^{1/2}$) for each sample[17]. Complete results can be found here: https://doi.org/10.7910/DVN/SJ8BEQ.

**Synchrotron-based micro-computed tomography (SR-μCT)**. Imaging was performed at the Biomedical Imaging and Therapy—Bending Magnet beamline (BMIT-BM 05B1-1) at the Canadian Light Source Synchrotron located in Saskatoon, SK. The beamline generates a continuous X-ray spectrum between 50 and 500 keV with the 12.6–40 keV energy range mostly used for experiments due to the beam size at full width half maximum of ~100.0 mm (H) × 3.2 mm (V). A 2.000 mm glassy carbon filter was used to generate a broadband beam with a mean photon energy of ~17 keV. A PCO Dimax (2000 × 2000 pixels) sCMOS detector was coupled with a Optique Peter (Lentilly) microscope (×10 objective, Mitutoyo LWD Plan Apochromat) to the Lutetium Silicon Oxide: Terbium (LSO:Tb) scintillator (10 μm thick, European Synchrotron Radiation Facility), which gave an effective pixel size of 1.2 μm with a field of view (FOV) of 2.44 mm².

Chocolate samples were cut into ~5 mm cubes and mounted on a Huber manual goniometer head with dental wax. The sample was placed 4.0 cm from the detector. Twenty flat images (no sample in beam path) and 20 dark images (no beam on the detector) were acquired in order to subtract the detector background noise and normalize the intensity of the X-ray images. Sets of 2000 projection images (sample in beam path) were collected through 180° rotation of the sample. All images were collected with an exposure time of 2 ms. Each set of 2000 projection images constituted one scan. To increase the total sample volume in the study, the sample was translated in the direction of the rotation axis (vertically) and two additional scans were collected. The volumes interrogated by the three scans had sufficient overlap to allow for the scans to be stitched together during data processing into one continuous cylindrical volume. This volume contained chocolate from the outer surface to a depth of ~6 mm.

Image processing and three-dimensional reconstructions were similar to those described by Willick et al.[21] In brief, the data were reconstructed using UFO-KIT software described by Vogelgesang et al.[22,23] (https://ufo.kit.edu/dis/index.php/software/ and https://github.com/sgasilov/ez_ufo) and the ezufo and ezstitch tools. A test image stack was generated in UFO and ring artifacts (concentric rings in reconstructed images) were suppressed using a low-pass Gaussian filter on the horizontal frequencies. Transport of intensity–phase retrieval (TIE-PR)[24] was performed on the test dataset utilizing the following parameters in the UFO-kit: Beam energy = 17.0 keV; pixel size = 1.2 μm; sample to detector distance = 4 cm;

$\delta/\beta = 200$. This test image stack was used to determine histogram clipping values for the conversion of 32-bit tiff image stacks to 16-bit tiff image stacks in the final reconstructions. ImageJ was then used to find the horizontal overlap of each view and EZ-stitch was used to generate orthogonal sections through the entire volume. The $450 \times 450 \times 450$ μm volumes were cropped from each CT scan using ImageJ. Final image stacks were opened, visualized, volume rendered and analyzed in 3D data visualization software Avizo 2020.2 (Thermo Scientific™ Amira-Avizo™ Software, Thermo Fisher Scientific, Waltham, MA USA). Volume renderings and orthogonal pan through of each sample can be seen here: https://doi.org/10.7910/DVN/WFUHDV. Histogram's were produced using excel, by plotting 16-bit bin data vs voxel count. Reconstructed slices used for analysis can be retrieved from https://doi.org/10.7910/DVN/G6QBD2.

**Statistical analysis**. GraphPad Prism software version 5.0 (La Jolla, CA, USA) was used for all statistical analysis and nonlinear regressions. The data collected was analyzed with one-way ANOVA tests considering $P < 0.05$ as significant with Tukey post-tests.

## Data availability

Source data for all figures and tables have been deposited in Harvard Dataverse (https://dataverse.harvard.edu) and the links indicated in the Experimental Procedures section of the manuscript.

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

## Acknowledgements
We acknowledge the financial support of the Natural Sciences and Engineering Research Council of Canada (NSERC) through grant RGPIN-04983, and the Canada Research Chairs (CRC) Program. Part of the research described in this paper was performed at the Canadian Light Source, a national research facility of the University of Saskatchewan, which is supported by the Canada Foundation for Innovation (CFI), the Natural Sciences and Engineering Research Council (NSERC), the National Research Council (NRC), the Canadian Institutes of Health Research (CIHR), the Government of Saskatchewan, and the University of Saskatchewan. CT data handling, processing, and analysis on this paper was partially supported by the grant and contribution-funding program of the National Research Council of Canada. J.A.S. would like to acknowledge Sergei Gasilov for technical support at the BMIT beamline and image reconstruction tools for CT data.

## Author contributions
J.C. contributed to the planning and execution of experimental procedures and the writing of the manuscript. S.M.G. contributed to the planning and execution of experimental procedures. J.A.S. carried out the synchrotron micro-CT data collection, analysis, and write-up. A.G.M. contributed to the experimental planning and the writing of the manuscript.

## Competing interests
The authors declare no competing interests.
