## [Peer Review File · Nature Communications]

REVIEWER COMMENTS

Reviewer #1 (Remarks to the Author):

The impact of tempering on the quality and storage stability of chocolate is a topic of longstanding interest, as is the use of additives to control the formation of fat bloom. Any method that simplifies the process of tempering is to be desired. The study of additives in the crystallization and polymorphism of fats also has a long history.

The present study is a very useful addition to the area and is certain to be of great interest to both fat researchers and chocolate producers; I would recommend publication following addressing of my comments below.

The methodology is described well and statistics used appropriately.

Expansion of some of the discussion would help to clarify things at certain points.

During the discussion of melting points and enthalpies (paragraph at lines 122-132), comparison is made to melting points reported by Wille and Lutton, and by Lovegren et al. However, the two groups used different methods, and different cocoa butters (and, consequently, reported slightly different melting points). It is, therefore, inappropriate to compare the melting point of the unrefined cocoa butter to that of Wille and Lutton's form VI melting point, but then to compare the melting points of the cocoa butter with added FFA to the form VI melting point of Lovegren et al. (the reported form VI melting points differing by around 3°C).

Since X-ray diffraction was performed at the final point of crystallization, no information on preceding polymorphic forms is available. Crystallization of cocoa butter at 5°C (whatever additive is present), will almost certainly lead to an initial formation of form III or IV. During holding at 20°C for 4 days, this will transform to form V. Supplementary figure 4 shows that form V exists in the recrystallized commercial chocolate as well as those with phospholipid added. Hence, polymorphically, the phospholipids have not had much impact. However, the hardness measurements do show a difference, which the authors rightly use as evidence that the impact on microstructure is key. In paragraph at lines 271-277, the authors mention that DMPC templates form V. This cannot be concluded from the evidence presented; templating implies the formation of a surface onto which the cocoa butter crystallizes immediately into a specific desired polymorph. Crystallization into the form V polymorph at 5°C has not been demonstrated. What can be implied by the evidence is that DMPC influences the initial nucleation and the subsequent recrystallization, impacting the final microstructure. This is exactly how a typical additive might be expected to function. Thus, the title of the manuscript should be altered to remove reference to molecular tempering, which has not been shown (time resolved X-ray diffraction during crystallization would be necessary to prove this).

Lastly, just a couple of notes on the Supplementary material. The fatty acid composition of GMO in Supplementary table 1 is missing around 20%, a significant amount of... something! It would be helpful,

in Supplementary figures 1 and 2, to indicate that the samples are cocoa butter with the various additives.

I hope this is useful & helpful!

Kevin Smith

Reviewer #2 (Remarks to the Author):

The major claim of the paper is that tempering of chocolate can be circumvented by adding minor lipid components. This aspect is novel although the results not sufficiently convincing. The relation between the results on cocoa butter (which are much less novel) and the results on chocolate are not clear enough. Results are of interest to the chocolate and fat crystallization community. Paper could influence thinking in the field if the results would be more convincing.

More detailed comments:

- line 47: do all authors agree that cocoa butter has at least six polymorphic forms?
- literature information about effect of minor components on cocoa butter crystallization does not seem to be complete, some references are present in the reference list but their content is not discussed in detail compared to other papers which are discussed in detail
- line 78: in materials and methods it says that RBD cocoa butter is used.
- crystal structure and polymorphism (lines 88 and following): (1) it should be mentioned here as well after which crystallization process, not only in M&M, I assume it is then static crystallization at 23°C for 1 day. (2) Is the fact that polymorph VI is then obtained in compliance with literature? (3) summary of information in table would be useful (4) is it not possible to roughly quantify the different polymorphic forms
- melting point and enthalpy of fusion (lines 122 and following): melting points depend very much on the procedure of measurement and the crystallization before melting thus care should be taken when comparing data with literature, same remark for enthalpies of fusion
- Line 131: why has the unrefined CB then a melting point of 36.4°C and a form VI?
- PLM: are the differences significant?
- Figure 4: given the figure, the R² value, taking into account the errors on the measurements, this correlation although statistically significant seems weak to me to draw strong scientific conclusions from it
- Line 205-206: this would be valid if form VI is formed directly from the melt, but this is not the case I assume
- Given the importance of the phospholipids I would use a more selective method than the determination of phosphorous content (LC-MS)
- Has there been any repetition of the refining and mixing process? Or has the correctness and

homogeneity of mixing been checked by chemical analyses?

- It is not mentioned where the added minor components were sourced from
- Why was a crystallization temperature of 23°C selected?
- Why was a heating rate of 5°C/ min selected in DSC?
- Has the quality of the results for surface colour obtained with a smartphone been validated with equipment more fit for the purpose?

REVIEWER COMMENTS

Reviewer #1 (Remarks to the Author):

During the discussion of melting points and enthalpies (paragraph at lines 122-132), comparison is made to melting points reported by Wille and Lutton, and by Lovegren et al. However, the two groups used different methods, and different cocoa butters (and, consequently, reported slightly different melting points). It is, therefore, inappropriate to compare the melting point of the unrefined cocoa butter to that of Wille and Lutton's form VI melting point, but then to compare the melting points of the cocoa butter with added FFA to the form VI melting point of Lovegren et al. (the reported form VI melting points differing by around 3°C).

Answer: Many environmental factors (temperature, rainfall, soil, and sunshine), agronomic practices (cocoa tree type, harvest period, and cocoa bean variety), post-harvest treatments, and CB refining can affect the functional properties of a given cocoa bean variety. However, considering all these factors, there is a limited, and well-defined, range of values for the melting point of refined and unrefined CBs in a specific crystal polymorphic form. The authors agree with the reviewer: Willie and Lutton reported a melting point of 33.8 °C and 36.3 °C for crystal forms V and VI, while Lovegren et al. reported a lower melting point, 30.0 °C and 33.3 °C, respectively. Our results agree with those of Willie and Lutton, so we have changed the discussion in the paper (highlighted yellow) to indicated agreement with Willie and Lutton and avoid comparisons to the work of Lovegren, which is outside the generally agreed-upon range. Thank you for pointing out this potentially confusing point.

Since X-ray diffraction was performed after crystallization was complete, no information on preceding polymorphic forms is available. Crystallization of cocoa butter at 5°C (whatever additive is present), will almost certainly lead to an initial formation of form III or IV. Upon storage at 20°C for 4 days, this will transform to form V. Supplementary figure 4 shows that form V exists in the recrystallized commercial chocolate as well as those with phospholipid added. Hence, polymorphically, the phospholipids have not had much impact. However, the hardness measurements do show a difference, which the authors rightly use as evidence that the impact on microstructure is key. In paragraph at lines 271-277, the authors mention that DMPC templates form V. This cannot be concluded from the evidence presented; templating implies the formation of a surface onto which the cocoa butter crystallizes immediately into a specific desired polymorph. Crystallization into the form V polymorph at 5°C has not been demonstrated. What can be implied by the evidence is that DMPC influences the initial nucleation and the subsequent recrystallization, impacting the final microstructure. This is exactly how a typical additive might be expected to function. Thus, the title of the manuscript should be altered to remove reference to molecular

tempering, which has not been shown (time resolved X-ray diffraction during crystallization would be necessary to prove this).

Answer: The authors generally agree about the minor components' effects as seed crystals formed during the initial CB crystallization stage. Comparing the crystallization growth curves of CB mixed with different minor components, we have shown that CB mixed with phosphatides, especially DMPC, had the shortest half-time of crystallization (112.3 min) compared to other samples. This proves that DMPC promoted crystallization of CB. In this study, the term "templating" was applied to explain the possible effects of phosphatides as an enhancer to promote CB's crystallization through heterogeneous nucleation. The crystallized DMPC molecules contain myristic acid as the main fatty acid, with a similar structure to saturated fatty acids in CB. The similarity in the 3-L structure of crystallized blend of CB and DMPC in the SAXS region demonstrates a similarity in lamellar structure (to native tempered cocoa butter), that was not observed with the other additives. However, we believe that our studies did not distinguish clearly between the templating effect of DMPC at a molecular level and its heterogeneous nucleation effect on CB crystallization. So, we have changed the title from "Molecular tempering...." to just "Tempering".

Lastly, just a couple of notes on the Supplementary material. The fatty acid composition of GMO in Supplementary table 1 is missing around 20%, a significant amount of... something!

Answer: The fatty acid composition of the GMO has been corrected in the supplement section.

It would be helpful, in Supplementary figures 1 and 2, to indicate that the samples are cocoa butter with the various additives.

Answer: The legends of both Figures 1 and 2 have been corrected as requested.

I hope this is useful & helpful!
Kevin Smith

Reviewer #2 (Remarks to the Author):

The major claim of the paper is that tempering of chocolate can be circumvented by adding minor lipid components. This aspect is novel although the results not sufficiently convincing. The relation between the results on cocoa butter (which are much less novel) and the results on chocolate are not clear enough. Results are of interest to the chocolate and fat crystallization community. Paper could influence thinking in the field if the results would be more convincing.

Answer: We clearly and statistically demonstrate the effects in cocoa butter and in chocolate made with that cocoa butter. We characterize solid state structure, all the way to mechanical properties and surface color. We have all the appropriate controls. We show "tempering" of cocoa butter without the usual physical tempering. I am not sure what else one would do to prove any phenomenon in nature.

More detailed comments:

- line 47: do all authors agree that cocoa butter has at least six polymorphic forms?

Answer: Most authors agree that six crystal polymorphic forms exist in cocoa butter based on over 80 years of published literature.

- literature information about effect of minor components on cocoa butter crystallization does not seem to be complete, some references are present in the reference list but their content is not discussed in detail compared to other papers which are discussed in detail.

Answer: The authors tried to address the related topic in the research to the selected references. For example, Professor Dimick has published many papers related to minor components' effects on CB crystallisation; however, two papers have been selected (reference 1 and 6) to discuss in detail. There are only a handful of studies on this topic to be found.

- line 78: in materials and methods it says that RBD cocoa butter is used.

Answer: It has been changed and corrected to CB (RBD eliminated).

- crystal structure and polymorphism (lines 88 and following): (1) it should be mentioned here as well after which crystallization process, not only in M&M, I assume it is then static crystallization at 23°C for 1 day. (2) Is the fact that polymorph VI is then obtained in compliance with literature? (3) summary of information in table would be useful (4) is it not possible to roughly quantify the different polymorphic forms

Answer: (1) The statement of "In this study, CB samples were melted at 80°C for 30 minutes to clear the memory of crystallization and then crystallized statically at 23°C in an incubator for 1 day." has been added to the crystal structure and polymorphism paragraph.

(2) The crystal polymorphic form VI was observed only for unrefined CB stored for a long time in the fridge.

(3) Supplementary Table 4, "Small-angle powder X-ray diffraction data for the CB in crystal polymorphic forms of IV, V and IV" has been added to the supplementary file.

(4) About the quantification of different polymorphic forms, since the coexistence of two polymorphic forms (for instance, form IV and V), the diffracted peaks related to each polymorphic form usually cover the other form. Also, in some cases, diffracted peaks are weak and wide, and it is not possible to get a rough area below the diffracted peak. Hence, the quantification of different polymorphic forms in a mix crystal structure is not feasible.

- melting point and enthalpy of fusion (lines 122 and following): melting points depend very much on the procedure of measurement and the crystallization before melting thus care should be taken when comparing data with literature, same remark for enthalpies of fusion

Answer: The authors agree with the reviewer; determination of melting profile using DSC analysis depends on many factors such as sample size, distribution of the sample in the DSC pan, heat transfer efficiency during heating, heating rate,...). However, we ran multiple replicates of each sample in identical fashion and compared them one relative to the others. We are confident in the accuracy of our results.

- Line 131: why has the unrefined CB then a melting point of 36.4°C and a form VI?

Answer: Since the CB was stored for a long period in the fridge, crystal polymorphic form was completely transformed to form VI with a melting point of 36.4 °C.

- PLM: are the differences significant?

Answer: Statistically, no significant differences ($P > 0.05$) existed.

- Figure 4: given the figure, the R² value, taking into account the errors on the measurements, this correlation although statistically significant seems weak to me to draw strong scientific conclusions from it

Answer: After plotting the fractal dimension's values as a function of melting temperature, we found a statistically significant relationship between the distribution and packing of crystallized mass and the melting point of crystals. However, we agree with the reviewer that the R square value of 0.6 is not very high, and it is not surprising because authors believe that many factors can affect the melting profile. However, all considered, we can explain 60% of the variability in fractal dimensions (i.e., microstructure) from the melting point of the CB crystals. Considering all, this is not too bad.

-Line 205-206: this would be valid if form VI is formed directly from the melt, but this is not the case I assume

Answer: That is entirely right. The CB in crystal form VI was only observed in crude CB that kept for a long time in the fridge. The CB in form VI can be obtained only through a polymorphic transition from form V.

- Given the importance of the phospholipids I would use a more selective method than the determination of phosphorous content (LC-MS)

Answer: The authors entirely agree with the reviewer; however, the AOCS Official Method has been selected as a more convenient method to analyze the phospholipid amount in samples.

- Has there been any repetition of the refining and mixing process? Or has the correctness and homogeneity of mixing been checked by chemical analyses?

Answer: The factors affecting the edible oil refining process (phosphoric acid, sodium hydroxide, water, and bleaching clay amounts, temperatures, and reaction times), was optimized over an entire year, and the efficiency of separation of by-products and quality of final products at each refining step was monitored by taking samples and checking the quality. Mixing temperature and time were also optimized for obtaining a homogenous mixture of bulk fat and minor components.

- It is not mentioned where the added minor components were sourced from

Answer: The sources of minor components have been added to the manuscript.

- Why was a crystallization temperature of 23°C selected?

Answer: We decided to crystallize our samples at ambient temperature, so 23 °C was selected as room temperature to crystallize our samples. Moreover, matching the crystallization results obtained at this temperature was easier to compare with previous studies in our laboratory (Campos et al. work).

- Why was a heating rate of 5°C/ min selected in DSC?

Answer: We decided to run the samples with the heat flow rate of 5 °C/min, because based on our previous experience, we found the rate of 5 °C/min, we can obtain sharp and separated peaks in the DSC thermograms. Moreover, the thermograms related to crystal polymorphic transition (the re-crystallization and melting) can be monitored using this rate. It is based on our 30 years of experience carrying out these measurements with these machines.

- Has the quality of the results for surface colour obtained with a smartphone been validated with equipment more fit for the purpose?

Answer: We have previously used a CM 3500-d spectrophotometer (Konica Minolta Sensing, Inc., Mahwah, NJ, USA) equipped with SpectraMagic NXCM-S 100 software to analyze the changes in the surface of CB after tempering. The smartphone app gave us similar relative results. These measurements are not absolute; they are good for relative comparisons, which was the purpose of this study.

REVIEWERS' COMMENTS

Reviewer #1 (Remarks to the Author):

My concerns with respect to the original manuscript have been addressed satisfactorily. I am very happy to see this paper published with these amendments.

Reviewer #2 (Remarks to the Author):

I find that the authors have not really answered all the questions raised by the reviewers and if they are answered in the rebuttal letter adaptations are often not made in the manuscript itself.

The authors often say they remain with their statement but don't really do a lot of effort in trying to make this clearer then. If the results are not sufficiently convincing for the reviewer and the authors think it is, maybe it has something to do with the way the paper is written. For me it was not the easiest to read.

Therefore for a high impact journal as nature Communications I still find this insufficient although the results are really of interest to the chocolate and fat crystallizing community and could influence the thinking in the field. So if more attention is paid to the comments of the reviewers and the results are explained better and thus more convincing, it still could be made acceptable

Response to Reviewers' comments

REVIEWERS' COMMENTS

Reviewer #1 (Remarks to the Author):

My concerns with respect to the original manuscript have been addressed satisfactorily. I am very happy to see this paper published with these amendments.

Answer: Thank you

Reviewer #2 (Remarks to the Author):

I find that the authors have not really answered all the questions raised by the reviewers and if they are answered in the rebuttal letter adaptations are often not made in the manuscript itself.

The authors often say they remain with their statement but don't really do a lot of effort in trying to make this clearer then. If the results are not sufficiently convincing for the reviewer and the authors think it is, maybe it has something to do with the way the paper is written. For me it was not the easiest to read.

Therefore for a high impact journal as nature Communications I still find this insufficient although the results are really of interest to the chocolate and fat crystallizing community and could influence the thinking in the field. So if more attention is paid to the comments of the reviewers and the results are explained better and thus more convincing, it still could be made acceptable

Answer: Thank you for the kind comments regarding the general interest of our work to the crystallization and Food Science community. We have corrected the manuscript to remove any confusing or ambiguous language and have even added a new dataset on synchrotron micro-CT imaging of chocolate microstructure. This shows again that the additives are helping temper the chocolate. We hope the effect is more convincing now.

REVIEWERS' COMMENTS

Reviewer #3 (Remarks to the Author):

I am reviewing the section of the manuscript on microCT.

The methods were in line with what is commonly done by other synchrotron microCT users. Having the link to the movies of slices and volume rendering is appreciated. As you'll see in my subsequent comments below, I think there are a few places where the authors should adjust their wording, but this does not fundamentally changes their conclusion.

The chosen visualization and plot/table for Figure 6 is probably not the most useful or sophisticated way to analyze this data, but it is not wrong. I would take a small issue with lines 284-285, where the authors state "The recrystallized Lindt sample displayed less low-density material compared to the fresh Lindt, DMPC and DPPE samples. This is indicated by reduced green voxels in the volume rendering data and the discontinuance of grey-values below ~20000 compared to the fresh Lindt, DMPC and DPPE samples, as seen in Figure 6."

The histogram does show that there is less low-density material, but the reduced green in the visualization is actually an indication of a reduction in high-density material, not a reduction in the amount of low-density material--this kind of volume rendering generally doesn't tell you anything about the low-density material because the low- and mid-density material are rendered as transparent. The histogram actually does show that for Lindt-R, there is a reduction in both the high and low values with an increase in the mid values. In other contexts I've seen this kind of shift in the histogram used to argue that the material is more mixed--when you have a wider histogram you have more separate materials, and then they combine and you have more of a material with an intermediate density. (I'm giving that for context, I'm not claiming that I know what is happening in this case).

Dula Parkinson, Advanced Light Source

RESPONSE TO REVIEWER COMMENTS

Reviewer #3 (Remarks to the Author):

I am reviewing the section of the manuscript on microCT.

The methods were in line with what is commonly done by other synchrotron microCT users. Having the link to the movies of slices and volume rendering is appreciated. As you'll see in my subsequent comments below, I think there are a few places where the authors should adjust their wording, but this does not fundamentally changes their conclusion.

The chosen visualization and plot/table for Figure 6 is probably not the most useful or sophisticated way to analyze this data, but it is not wrong. I would take a small issue with lines 284-285, where the authors state "The recrystallized Lindt sample displayed less low-density material compared to the fresh Lindt, DMPC and DPPE samples. This is indicated by reduced green voxels in the volume rendering data and the discontinuance of grey-values below ~20000 compared to the fresh Lindt, DMPC and DPPE samples, as seen in Figure 6."

The histogram does show that there is less low-density material, but the reduced green in the visualization is actually an indication of a reduction in high-density material, not a reduction in the amount of low-density material--this kind of volume rendering generally doesn't tell you anything about the low-density material because the low- and mid-density material are rendered as transparent. The histogram actually does show that for Lindt-R, there is a reduction in both the high and low values with an increase in the mid values. In other contexts I've seen this kind of shift in the histogram used to argue that the material is more mixed--when you have a wider histogram you have more separate materials, and then they combine and you have more of a material with an intermediate density. (I'm giving that for context, I'm not claiming that I know what is happening in this case).

Dula Parkinson, Advanced Light Source

RESPONSE:

We thank the reviewer for spending time on our work and for the useful comments. Our response is as follows:

The histogram false colouring settings were as follows:

Green: ~20,000 – 25,000 where pure green is 22,500
Yellow: ~25,000 - 35,000, where pure yellow is 30,000
Red: >35,000, where pure red is 35,000 and above.

All fractions below ~20,000 are rendered as transparent which would include any air bubbles. The transparency for the green composes the majority of the volume data which contain the bulk matrix/binding material of the chocolate, *i.e.*, the CB and fats. As such, the opacity setting of ~ 2 % is required to visualize the other internal components, *i.e.*, carbohydrate and fibrous cocoa solids and sugar crystals which are represented as yellow and red. Therefore, all solid material volumetric

information of the chocolate data is being visualized. Here 'low-density material' is being referred to as the CB/fats and can be misleading. We have clarified what is being visualized and removed term 'low density material'.